# From Active Participant to Active Researcher: What Do Young People Understand about Research?

**DOI:** 10.3390/children10061066

**Published:** 2023-06-15

**Authors:** Amelia Alias, Nurfaradilla Mohamad Nasri, Mohd Mahzan Awang

**Affiliations:** Faculty of Education, Universiti Kebangsaan Malaysia, Bangi 43600, Selangor, Malaysia; amelia.alias@ukm.edu.my (A.A.); mahzan@ukm.edu.my (M.M.A.)

**Keywords:** child participation, child as researcher, child councillors, child-led research, UNCRC

## Abstract

This paper discusses the outcomes of an initiative to empower young people as active researchers. It highlights participants’ understandings of their role as researchers in terms of the meaning of research, the research processes, and the participants’ competences, knowledge, and skills. It describes a process that a group of 15 child councillors aged between 15 and 17 years went through that was aimed at equipping them with the knowledge and skills to conduct their research. Based on the data presented, it was clear that from the participants’ point of view, the research process was challenging and time-consuming since it entailed several steps that needed to be carried out with careful attention to detail. Despite this, the participants were confident in their ability to undertake independent research, albeit with guidance from adults. Their research abilities and knowledge of child rights improved because of their having conducted research. The participants also noted that their learning was more meaningful when they were engaged in the subject topic, supporting the notion that learning by doing is vital. Due to its emphasis on acquiring children’s meaningful participation and illustrating the complex reality of being a part of research, this study has made important contributions to the small body of literature on child participation in Malaysia.

## 1. Introduction

Malaysia ratified the United Nations’ Convention on the Rights of the Child (UNCRC) on 28 December 1994 [1]. Since then, the country has committed to this code of obligations towards children. Among those responsibilities is increasing the awareness and participation of civil society, including children, in realizing children’s rights. However, as noted in our previous article, discussion and literature on children’s participation in the Malaysian context are scarce [2]. Because of this scarcity, children may not receive the attention they need, and this can have a negative impact on their ability to participate. Those in the Global North, such as in the European Union, recognise the value of involving children and young people in decision-making processes at every stage [3]. Efforts to promote community partnerships in the context of the Sustainable Development Goals are on the rise at the Association of Southeast Asian Nations (ASEAN). ASEAN emphasises the value of youth involvement in community recovery efforts following the spread of COVID-19 [4].

Children participating in research is not a new concept, especially in Western countries. Indeed, it has been common for some years now for children to conduct social research on their own, with support from adults acting as facilitators. Ergler [5] commented that studies on child-led research had only focused on the logistical challenges of payment, time commitment, quality of data, or future benefits for child researchers. He suggested that more emphasis should be placed on obtaining the meaningful participation of child researchers or the messy realities of becoming and being a researcher. 

With that in mind, the objective of the present qualitative research was to investigate, from the perspective of children, what they understood about research, their views on whether children could lead research, the benefits of child-led research, which research phase was the most challenging, and, finally, how their experiences as active participants helped while they conducted their own research. This article addresses such questions by drawing upon an empirical study of child-led research by the Petaling Jaya Child Councillors (PJCCs). This article begins by explaining laws and regulations related to children and the current situation on child participation in Malaysia. This is followed by an exploration of literature on children as researchers and contestations within the literature about the credibility of this idea in social research. This article then grounds the reader in various child–adult collaboration activities and a subsequent child-led research project. Based on these activities and research studies, this article presents a critical consideration about two questions: (a) How do young people understand research? (b) How had their experiences as active participants helped them to be child researchers?

## 2. Literature Review

### 2.1. Child Participation in Malaysia

Until now, Malaysia has maintained five (5) exceptions to the UNCRC, namely in Article (2) (discrimination), Article 7 (name and nationality), Article 14 (freedom of thought, conscience, and religion), Article 28(1)(a) (free and compulsory education at the primary level), and Article 37 (freedom from torture and other cruel, inhuman, and degrading treatment or punishment and arbitrary detention). The reason provided for Malaysia’s reservations to the provisions of the CRC is that they do not comply with the Constitution, national laws, and national policies of the Malaysian Government, which include Sharia law [6].

The Child Act of 2001 [Act 611], which was updated in 2016, is the country’s main law for protecting, caring for, and rehabilitating children. It can be said that the country’s focus is given more to child protection as compared to participation. The other key legislations related to children in Malaysia are the Domestic Violence Act (1994), the Evidence of Child Witness Act (2007), the Anti-Trafficking in Persons Act (2009), the Sexual Offences Against Children Act (2017), and the Penal Code. The Criminal Procedure Code and the Evidence of Child Witness Act are also critical components of Malaysia’s child protection framework. There were also a National Policy on Children (2009) and an associated Plan of Action, which focused on development and its relationship to children’s survival, protection, development, and participation. The National Child Protection Policy (2009) and its associated Plan of Action sought to ensure that children were protected from all forms of neglect, abuse, violence, and exploitation. However, both policies have lapsed.

In terms of children’s voices and participation, a survey carried out in 2017 by UNICEF [7] showed that the vast majority of children in Malaysia believed that their viewpoints were valued by their family, friends, and teachers. However, more than half (54%) of children in Malaysia believed that their voices were not heard at all by leaders, or that they did not contribute to the process of bringing about change. The overwhelming majority of respondents, 95%, said that global leaders should pay more attention to the concerns and ideas expressed by children and young people. This finding suggests that while there are some progressive mechanisms, there are also limits and deficiencies, as well as several hurdles, that prevent the effective and meaningful engagement of children and young people in the decision-making process.

Efforts have been made to uphold and increase children’s and adults’ understandings of the rights of children in Malaysia. Among them was the establishment of the Children Representative Council (MPKK) in 2011, wherein the newly created committee consisted of 30 young people aged 13–17 years from each state in Malaysia. However, due to the lack of information on the MPKK, it is impossible to evaluate the degree to which it encourages meaningful engagement for adolescents and the degree to which it is inclusive and representative of adolescents coming from marginalised and disadvantaged groups.

As a result of the changes that were made to the Child Act in 2016, the Ministry of Women, Family and Community Development established a National Council for Children. This council is charged with advocating on the behalf of children. Two of the council members are children who are part of the MPKK, and the purpose of the council is to encourage the participation of children in the decision-making processes related to issues that are relevant to them.

Another positive step for encouraging meaningful child participation in Malaysia was the appointment of the Children’s Commissioner within the Human Rights Commission of Malaysia (SUHAKAM). The primary responsibility of the Children’s Commissioner is to protect and promote the human rights of all children throughout Malaysia regardless of their social or economic statuses. In 2021, the Office of the Children’s Commissioner also established the Children’s Consultative Council, which consists of children and youth between the ages of 10 and 17, and its purpose is to serve as a platform for children’s thoughts and perspectives from across Malaysia as well as to advocate for children’s rights in Malaysia.

Efforts towards gaining the recognition of a Child-Friendly City Initiative (CFCI) organised by UNICEF are also seen as efforts to increase the participation rights of children in Malaysia. There is a growing interest among city councils to be part of this initiative in Malaysia. At this point, entities that have expressed interest in being part of the initiative include one candidate city (Petaling Jaya (PJ) City Council), six cities that have signed the memorandum of understanding in the state of Sarawak, and at least 23 city councils.

There have also been a number of conferences that facilitate child participation: for example, the ASEAN Children Forum, the “Be the Change. Speak up!” Children for Child Protection Forum in 2012, the Youth Forum ‘Children for a Better Digital World’ in 2015, and the Child Friendly Cities Children’s Conference in 2019.

Despite the existence of these forums, it is probable that there is a limited amount of meaningful child participation in Malaysia. The failure to make child participation a mandatory component in the main child-specific legislation, i.e., the Child Act 2001, could be one of the reasons that the efforts on child participation in Malaysia are not taken seriously [8]. The other critical gaps are the absence of national policies and guidelines on children’s participation, absence of data on children’s participation, absence of child participation assessment, and insufficient awareness programmes on children’s rights and participation [9].

Besides, although Malaysia has been part of the UNCRC for nearly three decades, in terms of the research conducted, there are still many gaps. Through the literature search conducted, it was found that the research on children’s rights in Malaysia has been more focused on legislation and child protection. It was found that studies involving the participation or voices of children and young people in Malaysia are still limited.

### 2.2. Children as Researchers—Definition, Benefits, and Challenges

There are a variety of approaches that have been used when evaluating children’s participation in research. Children have been seen variously in the study literature as objects, subjects, social actors, main participants, and co-researchers [10]. Kellet [11] presented study methodologies, identifying research on, about, with, and by children. Alderson [12] found that children take on the role of the researcher in three main ways. The first way is at school, where students often do study projects as part of their learning process without sharing the results or using them to push for change. The second way is when children take part in research, assisting adult researchers as “co-researchers”. The third way involves research conducted by children, frequently called “child-led research”. Child-led research is on the rise, and researchers and practitioners alike have been encouraging young people to take the lead on studies that directly affect their lives (e.g., Cuevas-Parra [13], Augsberger et al. [14], Anselma et al. [15], Graham et al. [16]).

The contributions of Kellett and colleagues [11,17,18,19] to the field of child-led research have been especially significant. In Kellett’s methodology, children are encouraged to determine their own priority foci, formulate questions and aims for their research, plan their investigations, create instruments for collecting data, examine their findings, and decide what to do next.

Child-led research is widely lauded for how it may improve the lives of the children who participate in it by respecting their rights, giving them a sense of agency, giving them opportunities to develop their talents, and making them feel that their contributions matter [19]. Graham and colleagues [16] reported that their participants had reflected upon the knowledge they had been gaining by participating in a pilot child-led research project by stating that it had offered them a chance to feel more empowered (because they could have more equal and respectful relationships with adults), have more social capital (because they could build new friendships and relationships), build and/or use social skills in a new and different setting, and feel more confident and secure in themselves. Kellet [18] argued that the most significant effect of research carried out by children was the contribution it provided to our awareness and comprehension of childhood and the environments in which children live. The generated data can be utilised using various methods: for instance, to increase awareness, knowledge, and comprehension and to provide evidence to support hypotheses. Occasionally, data pertaining to issues of political governance, environmental significance, or law can influence the formulation of policy. Children’s research can and should inform policy because it generates new information from children’s perspectives that adults may not be able to access.

However, the journey towards acknowledging children as researchers has not been without its challenges. Kellet [11] highlighted barriers and critical issues in acknowledging children as researchers, i.e., power issues between adults and children, children’s competences, perceived insufficient knowledge, lack of researcher skills, funding and resources, protection, and ethical responsibilities. Typical research-community scepticism towards child researchers is centred on the idea that they do not have the experience or maturity to conduct rigorous and valid studies. Child participation advocates disagree, arguing that children’s social experiences are more trustworthy indicators of ability [12,20,21]. Christensen and Prout [10] contend that children’s competence is different from that of adults, not ‘lesser,’ and that it should not be assessed using the same criteria. Although children may have ‘inferior’ knowledge to adults in many spheres of life, they have a greater understanding of childhood (in the sense of what it is like to be a child) [19]. James, Jenks, and Prout [22] stressed that children and young people were experts on their own lives and that therefore, research that drew on this knowledge would be more valuable if it was conducted in areas that were of direct interest to them; this argument has been concurred with by [12,13,16,19,21].

## 3. The Journey from Active Participant to Active Researcher

In ensuring the success of the Child Friendly Cities Initiative (CFCI), in October 2022, the PJ City Council appointed a group of school children aged 10 to 18 to join the child council. The purpose behind this child council is to provide a forum in which all children and adults may collaborate on becoming individually and collectively more influential and work towards meaningful child participation in Petaling Jaya. According to the claims made by Corsi [23], the recruitment in and utilisation of children’s councils constitute, by far, the most popular approaches that are being utilised to incorporate children in the process of local decision-making. Children’s councils provide opportunities for children to have their voices heard. Children and young people are given voices in the local government, and this enables them to participate; this is significant for them as citizens and helps them collaborate with various institutions. The child councils also become advocates for a wide range of programmes such as those that deal with the safety of children, their rights, and any other projects that have to do with children.

### 3.1. The PJCCs as Active Participants

To encourage active participation and to equip the PJCCs with knowledge and skills, and supporting the seventh (7th) principle of child participation [24], which states that adults should facilitate children’s participation effectively, various child-friendly programmes were conducted. Presented below are some of the impactful programmes conducted by the PJ City Council and attended by PJCC members:(a)Introduction to Child Rights and Child Participation

Article 42 of the UNCRC specifies that children should know about child rights. Considering this, this workshop aimed to familiarise children with the UNCRC, CFCI, and concepts of child participation and protection. A total of 44 children and young people attended a capacity-building session held for two days at the Petaling Jaya Community Library in October 2022. The workshop was designed to use an adult–child collaborative approach. It involved five PJCC alumni as child facilitators and five adult facilitators. The PJCC alumni had served as child councillors during the 2019–2022 term.

Note that the fifth principle of children’s participation [24] outlines that all methods used in dealing with children must be child friendly. Thus, every session was held in a relaxed and fun way. The session started with an “energiser” run by PJCC alumni to create friendly relationships between participants and facilitators. For this session, each PJCC was required to write about their appearance on a sheet of paper and then fold the paper into an aeroplane. They were then required to throw the folded paper. Each PJCC then needed to take a folded paper and guess the participants whose descriptions were contained in that piece of paper (Figure 1).

The first principle of child participation refers to being transparent and informative [20]. During this session, the PJCCs were introduced to the concept of the CFCI. They also learned about the role that UNICEF plays in CFCI implementation, other cities that had participated in the CFCI, and the path that the CFCI had taken in Petaling Jaya.

The next part of the workshop consisted of a sharing session on the UNCRC led by PJCC alumni. This session provided the PJCCs with an opportunity to learn about children’s rights and child participation and the significance of these concepts, which serve as the foundations of the CFCI. Their learnings included the ability to recognise the CRC in digital environments. A few multiple-choice questions were given as incentives for PJCC members to participate in this session. Motivation stickers were given out to PJCC members who had actively participated in the session (Figure 2).

To ensure that the PJCC members would gain a better understanding of children’s rights, they were divided into six groups. This session was assisted by adult facilitators. Next, the PJCCs were required to organise the CRC cards according to five goals, i.e., the right to be heard; right to be respected, appreciated, and treated fairly; right to social services; right to be safe; and right to family, play, and recreation. A discussion was then held with all the PJCC members wherein each group was required to state their opinions and reasons why they had put their cards under the above goals (Figure 3).

This other session that was conducted during this workshop was related to child protection. A Children’s Court Advisor who had extensive experience on issues and problems related to child protection shared their experiences of cases that had happened. The PJCC members then had a group discussion and made a presentation on child protection issues (Figure 4). In the final session, which was meant to bring out each PJCC’s creativity, they were asked to create a board game to promote the UNCRC (Figure 5).

(b)Workshop on the Development of a Child Participation and Protection Policy for the Petaling Jaya City Council

The purpose of the project was to acquaint and familiarise PJCCs with the ideas of an evidence-based policy, meaningful child participation, and intergenerational collaboration in the process of establishing a child participation and protection policy for the city. As part of the project, a workshop session that lasted for two days was held in December 2022. The first day of the workshop consisted of a consultation session with all the PJCCs and the second day of the workshop consisted of a consultation session with the adults who were members of the PJCC Technical Committee. The workshop was designed in a collaborative way, encouraging participation by the first author as the project coordinator, four lecturers from the Universiti Kebangsaan Malaysia (UKM) as adult facilitators, and fifteen PJCCs as child facilitators.

A pre-workshop session was conducted to inform the child facilitators on their roles, their meanings, the importance of policy, and how the workshop was going to be conducted. In this session, two of the PJCCs were assigned as the project lead and the assistant project lead. Their role was to explain, to the participants, the role that policy plays in organisation, and to lead the sessions. The other ten PJCCs served as child facilitators and conducted a group discussion with help from the adult facilitators. The child facilitators were given tips on how to navigate the discussion in an effective and ethical manner.

The consultation with children and young people was designed in a fun and engaging way wherein the concept of the World Café was used. The World Café approach is a participatory technique that moves participants beyond information transfer to information exchange by helping groups participate in constructive discourse, contributing to cognitive reframing and individual sensemaking, allowing for a different understanding of knowledge development, and moving participants beyond information transfer [25]. The participants were arranged in a cafeteria-style seating arrangement, with either four or five people seated at each table. At least three rounds of dialogue took place in a row, each lasting around 20 min. Each participant discussed one topic at each table at which they were seated. Following the conclusion of each round, the players stood up and went to new tables, where they resumed their conversations with new groups of people.

The topics of discussion were five: community and school, urban planning, events organised by the PJ City Council, sports programme, and policy and budget. Each topic represented one famous city in the world. The project lead and assistant served as pilot and co-pilot, respectively, while the adult facilitators acted as café owners and child facilitators as assistant café owners. The World Café approach was used to ensure that child participants were able to give opinions on each topic so as to ensure the richness of the data collected and to deter participants from getting bored. During the group discussion, the consultation focused on getting opinions and views on how the PJ City Council should include children as parts of its planning process in each topic area in a meaningful and safe manner (refer Figure 6 and Figure 7).

### 3.2. The PJCCs as Active Researchers

The PJCCs’ journeys as researchers started when they proposed to meet the Minister of Education to voice issues on education. They named their project ‘School Creates Future Leader: Is School Fun and Safe?’ The fourth principle of child participation states that the issues that will be discussed by children should have real relevance to their lives and enable them to draw on their knowledge, skills, and abilities [24]. Thus, the selection of this topic by the PJCCs was suitable and appropriate. This initiative must be considered as fully child-led research, as the first author served only as a facilitator to assist child researchers in conducting their research.

Shier [26] highlighted four specific things needed from adults who supported child researchers, saying that they must familiarise the latter with: (1) effective methodology that was suited to the experiences and abilities of child researchers, (2) skilled and sensitive process facilitation, (3) technical support, and (4) protection. In this workshop, most of these had been given earlier, as explained in Section 3.1, and while the PJCCs were doing research, as explained below.

(a)Workshops and Activities for Child Researchers

As Kellet [19] emphasised, it is crucial to provide children with quality research training to enable them to design viable research procedures that can withstand independent scrutiny. Model 3 in Kellet’s model of delivering research training, which divides training into various parts and delivers each part as a whole-day workshop away from a school environment, was used. Table 1 summarises the training sessions organised at different times between January 2023 and May 2023.

**Table 1 children-10-01066-t001:** Workshops and activities for child researchers.

Workshop Session	Topics	Details
1	Basic concepts in research	Discussing what research was, the research processes, data collection methods, data analysis, reporting, dissemination of data, the ethical aspect of conducting research, and researcher self-protection. Sharing information on Kim and colleagues’ [27] work “Developing Children as Researchers.”
2	Research design	Setting the research problems and objectives and determining data using focus group discussions (FGDs) and surveys as data collection methods (Figure 8). The research team suggested that the scope of child research should cover five topic areas: syllabus, mental health in school, students’ participation, protection in school, and facilities in the school. Setting that the focus of the study was for the secondary school level (for students aged 12 to 17 years).
3	Planning for data collection	Preparation for FGDs. Setting timeline for preparation of the survey questionnaire.
4	How to use Google Forms	Participants learned how to use Google Forms as tools for data collection.
5	Prepare questionnaire survey	Furnishing child researchers with research articles related to the five topics to help them come up with questions. The research crew was divided into three-person groups. The research team utilised Google Docs to create the list of questionnaire questions collaboratively. The questions were validated and checked by an expert (a lecturer from the Faculty of Education, UKM)
6	Reflection	What had the group gone through so far?

(b)Data Collection Session on the ‘Is School Fun and Safe?’ Study Project

The data collection session was organised and conducted by the PJCC researcher team at the open community centre in Petaling Jaya in late February 2023 once they had completed the 3rd workshop as explained in Table 1. The three-hour session was attended by around 30 schoolmates of the researcher team who were aged between 15 and 17 years old. Two collection approaches were used in this session: (1) FGDs and (2) simple surveys. The research team utilised both skills and experiences that they had (as explained in Section 3), as well as other programmes, in conducting this session: for example, they used the concepts of the World Café and having two child facilitators in each group. In the FGDs, the research team focused on gathering the problem, root cause, consequences, and recommendations on each topic identified (as mentioned in Table 1 above) (Figure 9). The survey questions were placed on a glass wall for participants to answer the provided questionnaire using dot stickers (Figure 10). Studies involving children require researchers to emphasise complex ethical considerations, such as informed consent, access, power imbalances, confidentiality, and protection [28]. To ensure that the session was conducted ethically, the team requested both parents and the participants themselves to sign consent forms before coming to the session. The participants were also briefed by the project lead regarding the matters of confidentiality, anonymity, and volunteering before the session started.

The research team also interviewed some participants to listen to their feedback about the session. All interviewees expressed appreciation for having attended this session and described the session as informative, fun, and engaging as it was organised by peers that had the same level of understanding as them and were easy to interact with.

## 4. Materials and Methods

### 4.1. Participants

Of the 44 who were currently PJCC for the period from 2022–2023, fifteen 15–17-year-olds were recruited, as they had spent most of their time participating in and organising child-led activities and research for the PJ City Council as explained in Section 3 above. The participants, who consisted of eight girls and seven boys, came from different backgrounds and socio-economic statuses and studied in separate secondary schools around Petaling Jaya. Information on the participants is shown in Table 2. In this study, these young researchers acted as research participants, not as researchers.

### 4.2. Research Methods

A qualitative approach was used to generate detailed data [29,30]. Three face-to-face FGDs, representing three topics of discussion were conducted; this process took about one-and-a-half hours for each session. In addition, a simple mini questionnaire was given to the participants to gather their consensus on the issues discussed during the FGDs (Figure 11).

As depicted in Table 3, some of the participants attended all the FGDs session and some of them did not due to their lack of availability to attend the sessions. The topics and number of participants for each focus group are shown in Table 3.

The focus group sessions were audio-recorded with the participants’ consents. To ensure credibility, consistency, and transferability, the member-check technique was applied by providing the participants with verbatim transcripts. The participants were given two weeks to read through the verbatim transcripts, and they were allowed to make inquiries. All of them indicated that they were satisfied with the substances of the verbatim transcripts.

The study was conducted in accordance with the Declaration of Helsinki, as required and approved by the Research and Ethics Committee at the Universiti Kebangsaan Malaysia (UKM). The UNICEF publication on Ethical Research Involving Children [31] was resorted to in order guarantee that the research was carried out in a manner that adhered to ethical standards. The publication emphasises the duties researchers have towards their participants, their colleagues, and themselves. The consent form was signed by everyone who took part in the current study. Before the session with the focus groups, it was vital to ensure that the participants understood the purpose of the research and were made aware that their participation in the study was entirely voluntary. The researcher gave the participants a concise explanation of the purpose of the study, what would take place, how long it would run, what was expected of them, the possible risks and repercussions of their participation, and how the data would be used.

In addition, the researcher explained the ethical concerns, reiterating their right to refuse to answer any questions and withdraw from the interview or focus groups at an time, and highlighted issues on confidentiality and anonymity. Alderson and Morrow [32] and Powell and Smith [33] suggest informing participants that their consent can be renegotiated. The researcher briefed the participants about the permission process, reviewed the form with them, and then asked for their signature before continuing with the focus group discussions.

### 4.3. Data Analysis

Thematic analysis was used when analysing the data collected during the focus groups. The approaches of topic (descriptive) coding and in vivo coding were used in order to identify both the themes as well as the subthemes. During the in vivo coding process, the actual words of the participants that came up in conversation were utilised to create the subthemes. During the descriptive coding process, a code was given to each paragraph on the basis of the issue that had been addressed (during the relevant focus group discussion) or what had been written [34,35].

## 5. Findings

Table 4 summarises the key themes and subthemes that were identified during the analysis of the data. The findings are presented based on three areas.

### 5.1. Theme 1: Young People’s Understanding of Research

#### 5.1.1. Subtheme 1: Research Is an Intensive Process

Based on the FGDs conducted, the participants were of the view that research is an intensive process, as expressed by B1 below:


*“The process of fully or intensively brainstorming a certain subject/topic, almost immersing yourself into that topic to the point that you can fully understand it and understand everything about that subject, every information, detail, and related factors. Then can use it to gain knowledge and back it with hard concrete evidence (can come in the form of collected data via questionnaire surveys, and opinions). In the end, it’s to establish new facts and ultimately come out with a solution.”*
 B1, FGD2. 

In a way, B1 wanted to convey that research was a complex process as it required an in-depth understanding of the topic and all related components in research and also required effort to collect data and finally establish new knowledge. The participants also understood that conducting research would take a long time as it involved various phases, as mentioned by G1:


*“Research takes time. I will relate to the project that we did—Is School Fun and Safe? I have to understand first the topic itself. Then, we need to break it into sub-topic, in my case, our group did School Syllabus. After that, we build questionnaires and conduct FGD to obtain data. Then will need to analyse that data. Then finally conclude the results and give recommendations. Furthermore, as we are students, we have other responsibilities as well.”*
G1, FGD2. 

#### 5.1.2. Subtheme 2: Qualities of a Good Researcher

Participants were also asked about the qualities of a good researcher. Enthusiasm, curiosity, open-mindedness, commitment, motivation, willingness to sacrifice, and an out-going personality echoed in the excerpts from two participants:


*“For me, no one is enthusiastic, because I believe that we need to be somewhat enthusiastic about the research topic that we are currently studying so that the level of interest that you have, will motivate us to try and produce good results. No two is curiosity. We need to ask questions to gain more understanding so that we can gain more knowledge and convert it to wisdom and that wisdom gained through successful research. No three is open-mindedness, treating every side equally, balancing out each answer, analysing the data and coming out with a conclusion.”*
B1, FGD2. 


*“Firstly, we must have the motivation and constant determination to make the research successful since research is a very long project, it’s not something that you can finish in less than a week. Some people when they do a project they don’t take into account the possible problems that could become the obstacles. Then they will get shocked and then the motivation will decline. If that is not consistent, then you will abandon the project. Secondly, we must be willing to sacrifice our time, energy and finance to garner and collect vital data to be converted to information for the research. Finally, should be somewhat of an outgoing person, who can easily socialize and interact with the targeted demographic of the research.”*
B2, FGD2. 

Another participant highlighted that as a researcher, one should be creative so that the approach used to collect data is fun and child-friendly, as expressed by the below excerpt:


*“Collecting data in a child-friendly way is important so that children [and young people] who become respondents will be more comfortable and have fun giving their views. Maybe the way we want to collect that data can hold an activity like the World Cafe concept using focus group discussion, similar like we did during the Policy Development Session. The way we collect the data is fun, where the emcees act as pilots and co-pilot and us being the café owners/facilitators. Respondents will be more interested in giving their views. So, for an adult researcher it needs to be someone creative and understand children [and young people].”*
G2, FGD2. 

This was supported by another participant who mentioned that a researcher should be someone who is supportive.


*“As researcher we should create a supportive environment so that the children [and young people] feel comfortable sharing their thoughts and experiences”*
G3, FDG2. 

#### 5.1.3. Subtheme 3: Child-Led Research Able to Produce Relevant Findings

Participants were asked about the benefits of child-led research. An interesting insight was provided by the participant below, who mentioned that the experiences of adults when they were young people might differ from those who were children at the present time.


*“The findings of this research will affect the young people themselves. So since it affects young people, then I think it’s best if young people play a key role—like a child-led project. So the advantage is that the data and information collected and analysed would be more relevant. So what is a better way to get data about young people, and the problems faced by young people- surely by the young people themselves right? Adults simply cannot become young people again because they cannot discard the adult baggage/experience they have acquired in the interim and will always operate and think through adult filters, even if these are subconscious filters”*
B2, FGD2. 

The second participant agreed with the remark that was made above. They also contrasted young people who had lived in various eras and discussed why it was meaningless to question adults even if adults had had the experience of being young people in the past.


*“Ya, I do agree with B2 that it would be unwise to try and apply principles of childhood from a generation ago to a contemporary childhood. Above all, we need to be able to learn and understand the lived experiences of children of today. Young people of the early 90s were placed in different situations compared to young people of the mid-2000s, especially due to the emergence of smart devices and the Internet. As such, problems then may currently have been resolved, yet novel problems emerge. So when you have young people who lived in the contemporary time, they are the ones who have the best opinions”*
B1, FGD2. 

The other participant highlighted that more in-depth and truthful data would be gathered if the data were obtained by child researchers themselves.


*“The engagement we had with other young people will be much deeper I guess in the sense we will have different data as compared to data collected by adults because we are closer, like they are our peers. They will be more honest with us compared to adults because we can understand each other because we are the same.”*
G1, FGD2. 

The mini questionnaire answers showed that all (13/13) participants strongly agreed that the findings of the research conducted by young people would be more meaningful if the research was about them and led by young people themselves as they were the experts in their own lives.

### 5.2. Theme 2: Young People’s Experience as Researchers

#### 5.2.1. Subtheme 2: Young People Capable of Doing Research

In one of the FGD sessions, participants were asked to give their opinions on adults’ perceptions of young people’s competences and lack of experience to conduct rigorous and valid studies. As shown in the excerpts below, both quoted participants were confident that young people could lead research and excel with adult facilitation.


*“Sceptics often argue that young people aren’t competent enough to engage in research and they’re too young and inexperienced. However, my belief is that young people’s competence is different from adults’. Adults cannot have the same engagement with other young people as their own peers. Yes, I think that young people are definitely more than capable of doing this. Take us, the PJCC as an example. We’ve conducted data collection during the policy development before and this really helped us in familiarising and what to expect during the process. In my opinion, with excellent guidance from adults, I am sure that young people can be nurtured to conduct research successfully.”*
G1, FGD2. 

Note that G1 related to the experience of conducting FGDs with young people, which she had gained during the previous activity. The same views were expressed by B1, who also emphasised the need for facilitation and that research should be truly child-led research without adults controlling projects.


*“I believe that every young people to a certain degree to a certain point and age can lead the research, but of course, we need a lot of guidance, not adults interfering, controlling or taking over, young people still able to lead the project with someone pushing us and give us some pointers and advice. In the end, we can slowly move forward and come out with successful research.”*
B1, FGD2. 

One of the questions asked in the mini questionnaire was about whether competence in young people was different from that in adults and not lesser and should not be assessed using the same criteria as those used for adult researchers. The other question was about whether participants were more comfortable communicating with peers and, therefore, whether studies by young people were able to obtain data that sometimes could not be gathered by adults. All participants (13/13) strongly agreed with both statements.

Participants were also asked if they thought that teaching young people about research through the concept of learning by doing was effective. Some responses from the participants were as follows:


*“Definitely! We learnt so much during the development of the policy event and it help us in conducting research.”*
B5, FGD1. 


*“Learning by doing allow us to gain experience, which is more meaningful for us.”*
B6, FGD1. 

One participant reflected on the way the presentation on child rights had been conducted by the alumni:


*“The explanation and activities we did about child rights on the first day we met was very helpful, but the presenter that day just read the slide. So may be need older children to present because I think they would be able to explain rather than read the slide.”*
G4, FGD1. 

#### 5.2.2. Subtheme 2: Require Adult Facilitation

As mentioned earlier, adult facilitation is important in helping young people lead research.


*“The help we need from adults to enable us to carry out the study better is with detailed guidance, how we want to start, what we need to do and so on, and adults also need to provide mental support to facilitate the management of the study that we will do and have fun with the study”*
G2, FGD2. 

G2 even mentioned that young people also needed mental health support as part of conducting the research process. On the other hand, the participant below mentioned the importance of adults accepting that young people were capable of leading their own research.


*“The help we need from adults to carry out this study well is the nature of openness among adults that young people can lead research. With this openness, it can generate two-way relationships between young people and adults. This is important because the relationship between these two parties can help visualize problem-solving and also provide a better learning situation at school than before.”*
B3, FGD2. 

B1 listed four criteria of a good adult facilitator as below, showing that an adult facilitator should be aware that his or her role as a facilitator was to be someone who understood young people, was willing to listen, was responsive, and had a sense of good facilitating skills.


*“Firstly, an easy-going person, where he or she can talk to children in a relaxing/casual way and not so much formally. Secondly, patient and understanding—must understand that not all children are as intelligent. Third, open-minded—willing to listen to what children have to say and then respond to them. Fourth, flexible where he or she can adapt to each child’s ability in the group. Finally, great management. Someone who can facilitate group and stay on track to achieve the goal and able to manage group dynamic.”*
B1, FGD2. 

#### 5.2.3. Subtheme 3: Data Collection Is the Most Challenging Phase

The participants were asked to reflect on the research processes that they had gone through so far. One of the questions asked the participants what would happen if the World Café concept was not used in collecting data. One participant said:


*“I think we won’t be able to get much data from child participants. Participants might be bored because they will have to stay to discuss the same topic. With the World Café concept, participants will have to move around and give opinions on various topics.”*
B5, FGD2. 

Opinions from the participants were also sought to ascertain which research phases were most challenging to them.


*“I’d say that the toughest phase would be collecting the data. To do that we have to conduct physical face-to-face discussions or online surveys. On paper, these do sound simple, but in actual reality and practice, the process of conducting and preparing these discussions and surveys can be very challenging; require a lot of effort (logistically and financially), time and energy.”*
B2, FGD2. 

The same views were expressed by another participant:


*“I feel that collecting the data has to be the most challenging phase. Just like my friend mentioned, physical face-to-face discussions require a lot of energy and understanding. Making sure the participants understand why they are here, how they are making an impact on this research and conducting the discussion on certain topics is pretty difficult to do in a limited amount of time.”*
G1, FGD2. 

The other participant, however, related the conduction of literature reviews as challenging since it required researchers to do a lot of reading.


*“I think comparing articles and studies is quite a boring and annoying process because you have to go through so many articles and read each one and sometimes it does not really highlight the key points, which is even more annoying to me.”*
B1, FGD2. 

### 5.3. Theme 3: Benefits of Child-Led Research

#### 5.3.1. Subtheme 1: Child-Led Research Increased Understanding of Child Rights

As expected, research had increased the participants’ understanding of child rights. This was explained by the below-cited participants.


*“My understanding of children’s rights has greatly improved since the activities and research I participated in allowed me to be more familiar with the topic. For example, the research we are working on (“Is school fun & safe”) can be related to Article 12 of the rights of a child. Said article emphasizes the importance of involving children in decision-making processes that affect their lives. In the context of how to improve a school, conducting research gives us a better understanding and teaches us to practice our rights as a child. At the same time, it can help ensure that our perspectives and needs are being heard and taken into consideration.”*
B4, FGD3. 

A similar but more detailed explanation was given by B5, as below:


*“First, the research that we conducted is about school, so it falls under the rights to education. Then, the topic itself asked whether the school is fun and safe, so it covers protection rights. The subtopic that we want to explore further also involves child protection—under the school facilities and mental health topic, child participation and voices at school and, finally, rights to access information through the topic syllabus that we are going to focus on. So focusing and delving into the subtopic itself could further strengthen our understanding of child rights.”*
B5, FGD3. 

The participants were also asked to reflect on the current condition of awareness on child rights at school. The below was the typical answer by the participants.


*“In my experience, the discussions on children’s rights in school are quite limited and don’t capture my attention. These topics receive limited attention and are not given the depth of understanding they deserve.”*
B4, FGD3. 

The same view was echoed by B1:


*“To be honest, my school has not taught us about basic children’s rights. When it comes to the rights to be safe, the rights to be heard, and the right to voice out, I feel that those rights are implemented at school but the scope is very limited. For example, the rights to be safe, school to ensure our safety or the right to be heard, and the right to voice out, they allowed us to do so, despite the very limited scope and they never taught us about any of those rights. I think in my school since I am the PJCC, I’m the only one that knows about child rights, but my friends and the other students, do not have a clue about it.”*
B1, FGD3. 

The answers to the mini questionnaire also revealed the same findings: a majority (10/13) of participants strongly agreed that their school had not emphasised child rights lessons.

#### 5.3.2. Subtheme 2: Child-Led Research Increased Skills Needed for the 21st-Century

The data obtained from this study also showed that not only the participants’ understandings of child rights, but also each participant’s confidence, self-esteem, and decision-making skills had increased after they conducted their research.


*“The experience of participating as active researchers is an empowering process that leads to a virtuous circle of increased confidence and raised self-esteem, resulting in more active participation by young people in other aspects affecting their lives”*
B2, FGD3. 


*“The process of identifying problems can involve young people in the decision-making process, and this process gives young people a voice in deciding as well as shaping their research.”*
B4, FGD3. 

Another participant related the autonomy he had had while conducting research. Note that 21st-century learning encourages students to take charge of their own learning, thus enabling them to act in learning situations.


*“For me, I got to fight for the problem I faced in school or as a student. I am not researching super-physics, thermo dynamics or what so ever. So I am very familiar with my problems, except I do have to go a bit more in-depth, so the advantage to me is that I would be able to fight for that without too much adult interference, I have total control at the same time I am still guided, so my research could be validated. Another advantage is that I gain a new experience. It’s more important to us because we are advancing a much faster phase, and by the time we reach university level, we will be further advanced because of these early experiences.”*
B1, FGD3. 

The mini questionnaire asked the participants whether they agreed that their experiences as researchers or participants in attending the activities and conducting the study ‘School Creates Future Leader: Is School Fun and Safe?’ had really helped them in improving skills required for students to excel. As shown in Table 5, most of the participants thought that the experience they had received by participating actively in activities and research had assisted them in enhancing abilities necessary for the educational environments of the 21st century.

Understanding that conducting research helped students prepare themselves to be ready for the 21st century learning environment, participants were asked whether school emphasised on this fact.


*“Honestly, my school didn’t really help me to become a researcher, but MBPJ’s activities helped me.”*
G2, FGD2. 


*“I think the school did not make us become researchers unless there is a special project for certain subject like history, accounting, science computer, we do research on one particular topic or chapter that is asked, but apart from that we do not do any kind of social research. We do not research anything else apart from the topic given, no topic like child rights, child protection or participation like that. I feel MBPJ has taken a good initiative for us to do research and learn how to do research, what to do and how and what to find.”*
G4, FGD2. 

The mini questionnaire answers showed that 9 out of 13 participants said school did not expose students to projects involving community, while 10 out of 13 participants strongly agreed that the reflection process, which was part of conducting research, would help students improve the critical thinking skills.

Overall, the findings highlighted mirror those of previous studies that examined issues and debates on young people as researchers, as discussed in Section 2. These findings have shown that young people are confident that they are capable of taking control of research with adult facilitation and training. They understand that conducting research requires dedication and willingness to sacrifice time and energy. To be a good researcher, ones should be open-minded and have curiosity and enthusiasm since research will take a long time to be conducted. The findings also show that the participants concur that child-led research can produce in-depth, truthful, and meaningful data that an adult researcher may not be able to gather. Based on their experiences conducting research, the participants feel that the data collection phase is the most challenging in research. The findings also demonstrate that child-led research increased their understanding of child rights and increased their proficiency in skills needed for 21st-century environments. Finally, the findings of this study have highlighted that learning in school currently does not emphasise child rights or facilitate students to be good researchers.

## 6. Discussion

The purpose of this research was to investigate how young people perceived research and how their involvement in the process affected the quality of their findings. In other words, the study aimed to provide empirical evidence on child-led participation by children and young people in Malaysia, which is scarce in the research literature.

The findings in Theme 1 have shown that the participants understood research as a complex process that would take a long time to be done. Note that participant G1 related this understanding with the experience she had in conducting research, indicating that she had a good understanding of the basics of what research was. The participants also listed down the qualities of a good researcher, where five out of ten coincide with the criteria listed by Toledo [36]. Interestingly, the participants highlighted that a researcher should be someone who is creative and able to understand children and young people in order for them to feel comfortable and willing to participate during data collection. Note that participant B1 highlighted that a researcher should be open-minded in the sense of not being biased. Studies that include children and young people need researchers to give attention to a plethora of complicated ethical concerns. These issues include informed consent, access, power inequality, confidentiality, and protection [28,37]. Hence, although it was not brought up explicitly by any of the participants in this study, it is important to point out that one of the qualities a researcher needs to possess is a sense of ethics, in the sense that they respect human dignity, privacy, and autonomy, use extra care around vulnerable groups, and make an effort to properly divide the rewards and responsibilities of research. The findings in Theme 1 have also highlighted that research conducted by young people is able to produce relevant findings; this has been stressed by many child-led-research advocates [12,13,16,19,21]. It is even more intriguing that participants B1 and B2 were able to discuss and compare childhood circumstances, highlighting the fact that adults would think differently from them even if they had been children once and that certain knowledge may not be relevant in the present environment because of several changes.

The participants’ experiences were considered essential for this study and were discussed in Theme 2 of this study. Despite their understanding that research was complex, participants expressed their confidence that they could take control of research with adult facilitation and training. This finding is in line with those of studies by Graham and colleagues [16] and Shier [26]. The participants also highlighted that learning by doing, a concept introduced by John Dewey, is the best approach while teaching children about research.

The importance of the role of adults as facilitators who provide logistical support and provide advice so that child researchers are able to decide and carry out research independently was a common view expressed by the participants in this study. Ergler [5], Cuevas-Parra [13], and Le Borgne and Tisdall [38] highlighted the importance of the emotive elements of trust and relationships between adult facilitators and child researchers to ensure the success of research. Based on experience facilitating participants for the conduction of their own research, the first author concurs with this. In addition, an adult facilitator must be aware of the schedules of children and young people so that they may avoid scheduling any events during weeks of exams or during schooldays. More importantly, the adult facilitator should be aware of the strengths and weaknesses of children and young people to be able to manage group dynamics. Finally, the findings in Theme 2 highlight the participants’ view that data collection is the most challenging due to the limitations they face as students in term of time, finances, logistics, and energy. Vast studies concur with this notion that data collection is the most challenging phase in research [39].

The findings in Theme 3 highlight the benefits that the participants gained as researchers in terms of deepening their understandings of child rights. Moreover, each participant’s confidence, self-esteem, and decision-making skills also increased as they actively participated in activities organised by the PJ City Council and conducted research. This finding further strengthens the earlier claim made by Kellet [17], Graham and colleagues [16], and Kim and colleagues [27]. Note that one participant mentioned having autonomy while conducting research. Howe and Strauss [40] highlight the significance of allowing children to obtain their autonomy as this encourages children to develop intrinsic motivation to develop critical thinking skills and take ownership of their life skills and learning, both of which are essential attributes for 21st-century learning. Through self-directed learning, the 21st-century learning environment emphasises providing children and young people with the autonomy to be responsible and take charge of their own education [41]. An interesting finding highlighted by the participants is that their schools do not expose them to projects involving the community. Community services are known for bringing benefits to students as they increase social awareness and responsibility, boost self-confidence, self-esteem, and sense of accomplishment, help students stay physically active, and improve future job prospects—all requirements needed to survive in the 21st-century environment.

The findings also highlighted that child rights awareness programmes for students are still scarce. As expressed by the participants, the discussion on child rights and the implementation of child participation, especially in voicing out opinions, is limited at school. This is true in the sense that there are not many schools with a students’ association, committee, or union, students are not consulted regarding school regulations and policies, and students have limited and unsafe channels for voicing their opinions and complaint mechanisms for addressing bullying. Children and young people are often afraid to express their opinions to teachers and rarely disagree with them. The teacher-centred learning approach that has been in existence for a long time in Malaysia—especially in public schools—does not encourage open discussions. Therefore, critical thinking is not encouraged. This has led to children and young people experiencing a lack of confidence, being shy in the school environment, and being afraid of being scolded or mocked by teachers and peers. Consequently, this may have implications for the future civic participation of children and young people. These traits are common among Malaysian children and young people, especially those from poor and low-income families. The failure to make child participation a mandatory component in the main child-specific legislation, i.e., the Child Act 2001, could be one of the reasons that the efforts on child participation in Malaysia are not taken seriously [8].

## 7. Conclusions

Taken as a whole, the findings of this study highlight young people’s understanding of research as having the sense that the research process is complex and requires time to be completed as it involves various phases and attention to detail. Despite this, the participants are confident to conduct their own research but require adult facilitation. Conducting research increased their understanding of child rights and enhanced their soft skills. More importantly, the participants expressed that learning by doing was important as it encouraged meaningful learning, thus enabling them to retain more information when they were interested in the subject matter.

This study makes a significant contribution to the limited literature on child-led research and child-led participation by going in-depth, from young people’s perspectives, regarding how they understand research while conducting and leading their own research. It highlights young people’s views on how research conducted by them could contribute to the relevancy and truthfulness of data that adult researchers may not be able to obtain. This study has directly explored the claims made by previous authors that child-led research benefits children and young people in terms of deepening their understanding of child rights and enhancing their learning skills to enable them to embrace the 21st-century learning environment. The other contribution of this article is the detailed explanation, in Section 3, of how children’s rights awareness activities can be done in a child-friendly, ethical, and meaningful way with the involvement of children and young people as the main facilitators and planners of such activities. This may be able to be a guide for city councils in Malaysia that are planning to get CFCI recognition from UNICEF.

One limitation in this study is that at the point at which this study was conducted, the research on the project titled ‘Is School Fun and Safe?’, which is led by participants, was still ongoing. It is anticipated that more insightful data can be obtained if participants complete the entire research process up until the end, although a longer time will be taken. Another limitation is that only young people aged 15 to 17 took part in this study. Even though young people may have a good idea of how important research is, it would be helpful to have conversations with much younger Child Councillors to find out what they know about research. This study might have benefited if various data collection methods had been utilised: for example, the young people’s research journal diary approach, where more in-depth and meaningful data would have been obtained. Finally, future studies could investigate how far child-led research as part of project-based learning has been implemented in the educational context of Malaysia. Here, the theory of constructivism, which was absent in the currently reported study, could then be applied to support the concept of ‘learn by doing’ or learning through experience by children and young people.

## Figures and Tables

**Figure 1 children-10-01066-f001:**
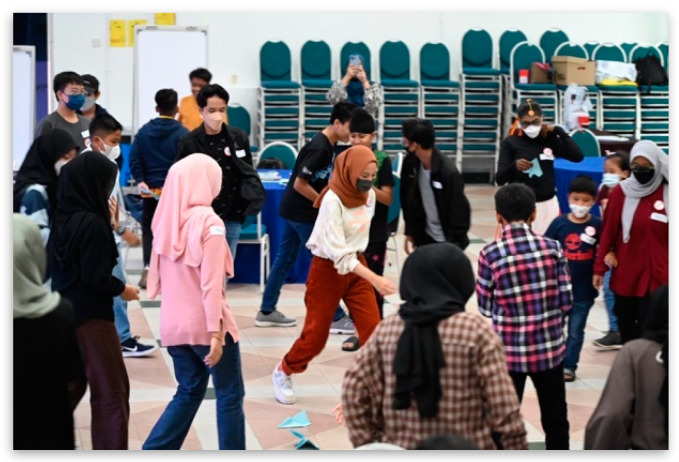
Energiser session. Source: first author collection.

**Figure 2 children-10-01066-f002:**
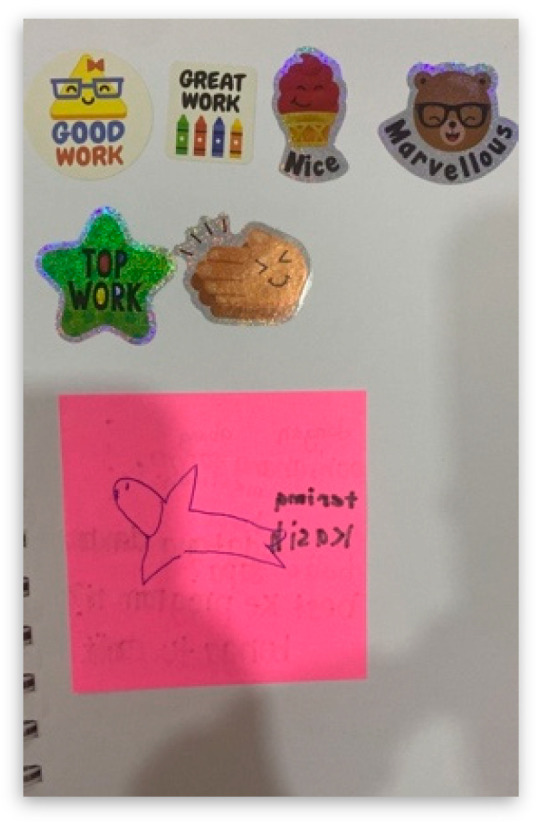
Motivation sticker. Source: first author collection.

**Figure 3 children-10-01066-f003:**
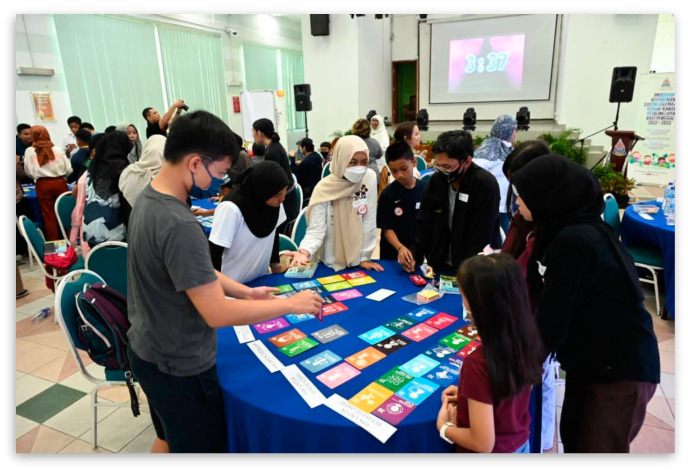
Group discussion. Source: first author collection.

**Figure 4 children-10-01066-f004:**
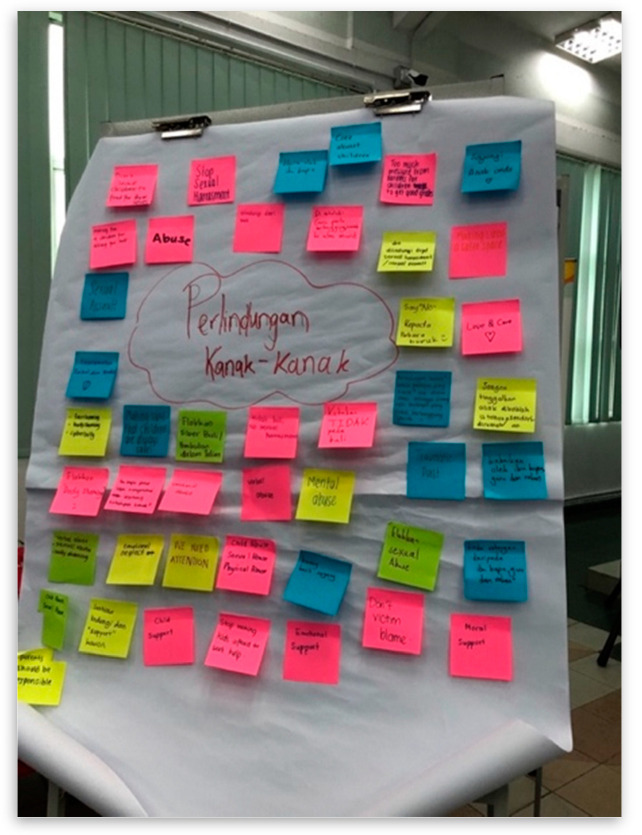
Children and young people’s opinions and views on child protection. Source: first author collection.

**Figure 5 children-10-01066-f005:**
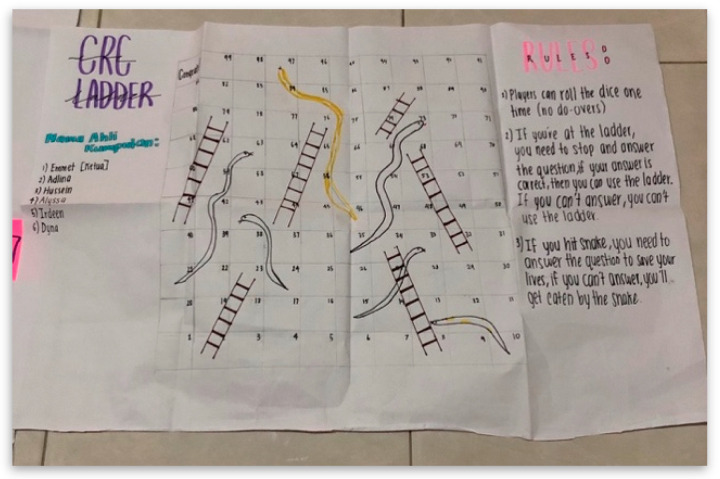
CRC snake and ladder game. Source: first author collection.

**Figure 6 children-10-01066-f006:**
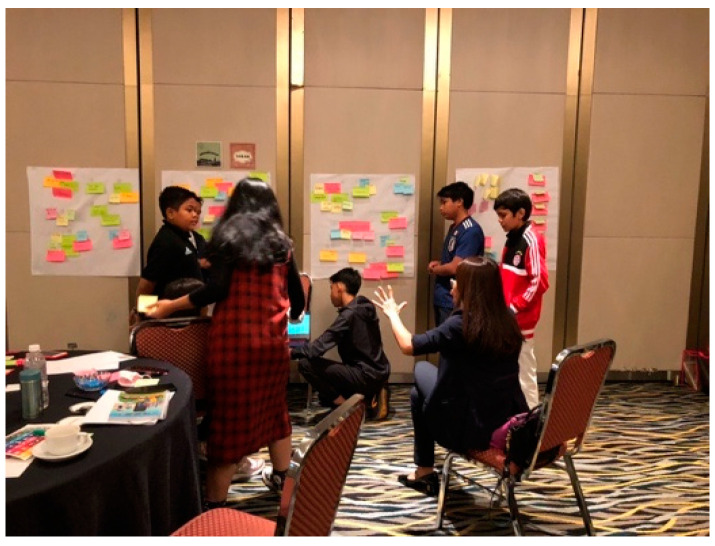
Consultation with children sports programme. Source: first author collection.

**Figure 7 children-10-01066-f007:**
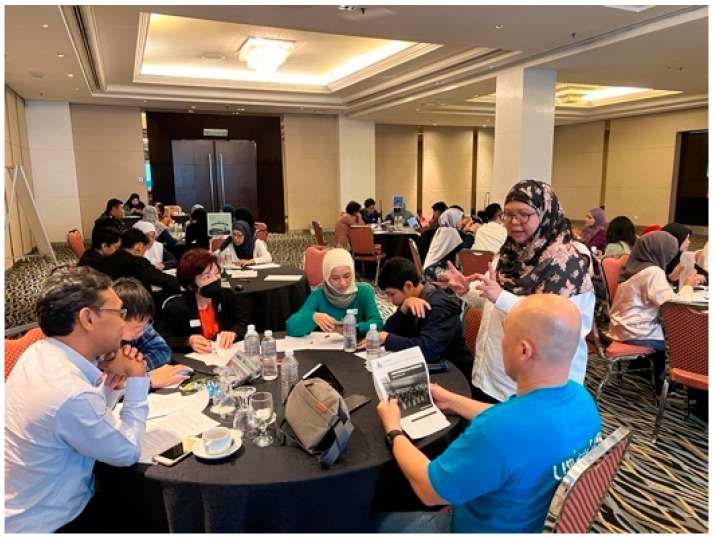
First author with representative from UNICEF Malaysia on the 2nd day’s intergenerational session. Source: first author collection.

**Figure 8 children-10-01066-f008:**
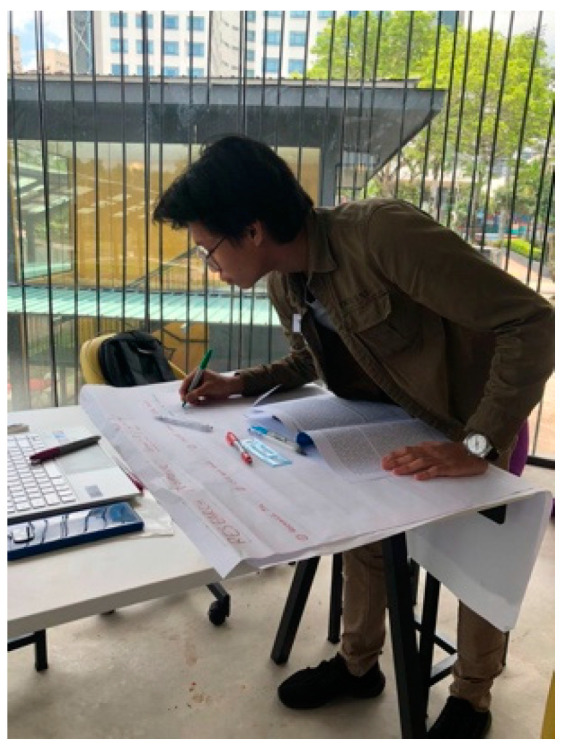
Project team lead identifying research problems and objectives. Source: first author collection.

**Figure 9 children-10-01066-f009:**
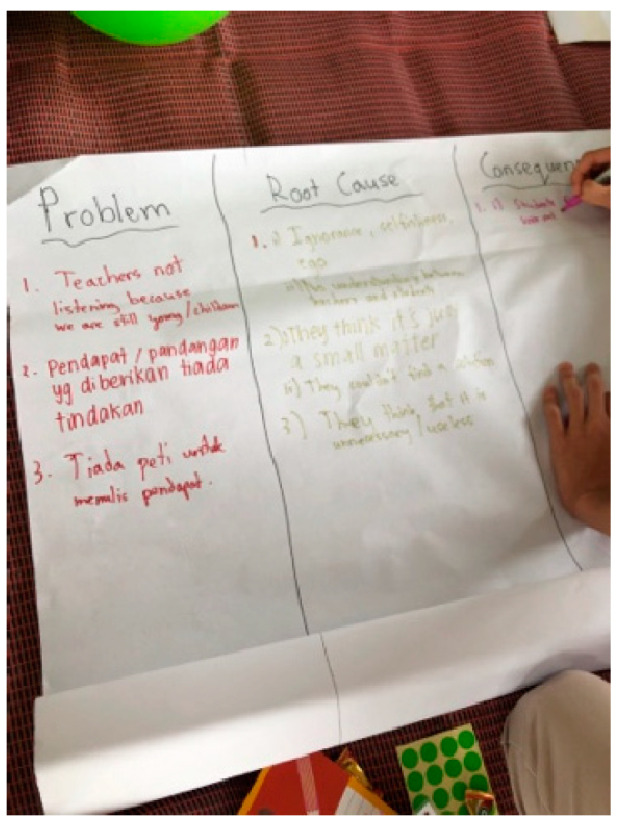
FGD session. Source: first author collection.

**Figure 10 children-10-01066-f010:**
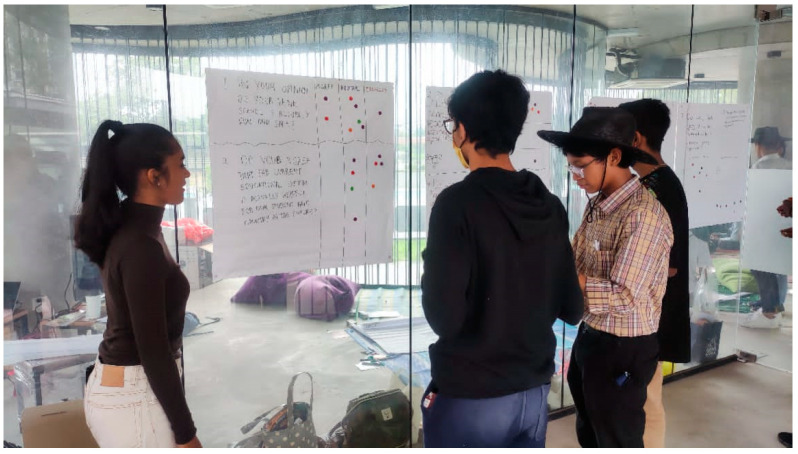
Survey gallery. Source: first author collection.

**Figure 11 children-10-01066-f011:**
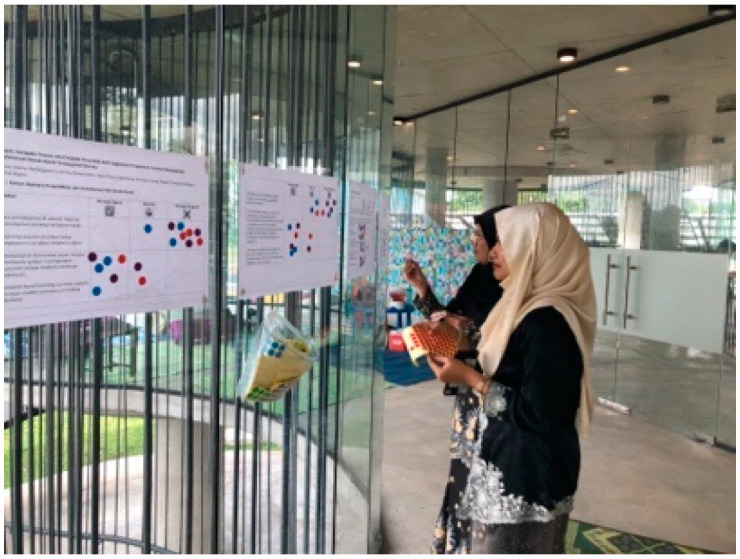
Participants answering the mini questionnaire. Source: first author collection.

**Table 2 children-10-01066-t002:** Participants’ information.

Girls’ Participants	Participants’ Age	Boys’ Participants	Participants’ Age
G1	16	B1	16
G2	16	B2	16
G3	16	B3	16
G4	16	B4	16
G5	16	B5	15
G6	16	B6	15
G7	15	B7	17
G8	15		

**Table 3 children-10-01066-t003:** Focus Group Interview Questions Guideline.

Focus Group Session	No of Participants	Topics
1 (FGD1)	8	Reflection on programmes attended and organised
2 (FGD2)	8	PJCC as Researcher—The Process
3 (FGD3)	13	Relationship between research and child rights

**Table 4 children-10-01066-t004:** Emerging findings based on analysis. Source: data from focus group discussion.

Theme	Subtheme
Young people’s understanding of research	Research is an intensive process Qualities of a good researchers Child-led research able to produce relevant findings
Young people’s experience as researchers	Young people capable of doing research Require adult facilitation Data collection is the most challenging phase
Benefits of child-led research	Child-led research increased understanding of child rights Child-led research increased skills needed for the 21st-century

**Table 5 children-10-01066-t005:** Participants’ mini questionnaire answers.

21st-Century Criteria	Strongly Agree	Neutral	Strongly Disagree
More creative	13	0	0
More IT literate and know more about information security	12	0	1
Better communication skill	13	0	0
More skilled in group work	13	0	0
More empathy	10	2	1
More confident as students	11	2	0
Improve learning skill	11	2	0
Improve problem solving skill	13	0	0
More knowledgeable about problems involving young people	13	0	0

## Data Availability

The data presented in this study are available on request from the corresponding author.

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
