# Peer review of "From Active Participant to Active Researcher: What Do Young People Understand about Research?"

_children, 2023, doi:10.3390/children10061066_

Round 1

Reviewer 1 Report

Thank you for the opportunity to review this article. I liked the authors' idea of structuring the text in a way that shows it as a journey and also provides visual material. In qualitative research this is an acceptable solution. However, it would be advisable to reinforce the article with scholarly sources, as some of what is currently available are documents. I also don't understand why the authors choose the term children if the participants are aged 15-17. I think this may be confusing for the reader I would suggest changing it to teenagers. Finally, the literature references (e.g. 8 and 18) are incorrect. It is recommended that the authors review the reference list and clarify the bibliography of sources.

Author Response

Thank you for being so helpful in reviewing this manuscript. Please review the response attached. 

Reviewer 2 Report

The article is focused on the critical topic of involving young children in research activities. The scope of this article is good, but it requires some minor corrections before it is published. 

1. The introduction seems fine, but it is limited to providing the main background information on the importance of the topic in international and regional contexts. 

2. The literature review covers the main challenges and opportunities for children to be researchers. A subheading is not required if there is only one. Repetition of words is present, for example, in the second paragraph, "Children are children are " repeated. 

3. This study used Article 42 of the United Nations Convention on the Rights of the Children to explain the right of children. It would have been interesting for the reader to learn the regional laws and policies for children's rights in Malaysia. A few lines on regional and national laws would be helpful for readers. 

4. Ethical considerations should be followed for all figures, including real human beings, in the article. Since the participants were minors (below 18), regional and international protocols should be followed for using human participants in research. A Snapshot of the social media page should also be checked if it ensures the anonymous identity of the participants. The consent statements should be reviewed, can minors consent for their participation in research? Children | Instructions for Authors (mdpi.com)

5. Participants' background information is not provided. It would be difficult for readers to understand the results, like who G1 is, as there were 8 participants in the first focus group discussion. It can be done by adding a table including background information on all participants.

6. The study limitations in the last section are not very well explained, and the study has methodological and theoretical limitations that should be described. 

7. Language of the article is satisfactory. However, proof-read by an expert in academic English is required. 

Author Response

(The authors gave the same response as above.)
